# AcrIIA5 Suppresses Base Editors and Reduces Their Off-Target Effects

**DOI:** 10.3390/cells9081786

**Published:** 2020-07-27

**Authors:** Mingming Liang, Tingting Sui, Zhiquan Liu, Mao Chen, Hongmei Liu, Huanhuan Shan, Liangxue Lai, Zhanjun Li

**Affiliations:** 1Key Laboratory of Zoonosis Research, Ministry of Education, Jilin University, Changchun 130062, China; liangming0107@outlook.com (M.L.); suitingting@jlu.edu.cn (T.S.); liuzq9914@mails.jlu.edu.cn (Z.L.); chenmao18@mails.jlu.edu.cn (M.C.); lhm2017851002@163.com (H.L.); shanhh18@mails.jlu.edu.cn (H.S.); 2CAS Key Laboratory of Regenerative Biology, Guangdong Provincial Key Laboratory of Stem Cell and Regenerative Medicine, South China Institute for Stem Cell Biology and Regenerative Medicine, Guangzhou Institutes of Biomedicine and Health, Chinese Academy of Sciences, Guangzhou 510530, China; 3Guangzhou Regenerative Medicine and Health Guang Dong Laboratory (GRMH-GDL), Guangzhou 510005, China; 4Institute for Stem Cell and Regeneration, Chinese Academy of Sciences, Beijing 100101, China

**Keywords:** base editor, Anti-CRISPR, off-target effects, mammalian cells

## Abstract

The CRISPR/nCas9-based cytosine base editors (CBEs) and adenine base editors (ABEs) are capable of catalyzing C•G to T•A or A•T to G•C conversions, respectively, and have become new, powerful tools for achieving precise genetic changes in a wide range of organisms. These base editors hold great promise for correcting pathogenic mutations and for being used for therapeutic applications. However, the recognition of cognate DNA sequences near their target sites can cause severe off-target effects that greatly limit their clinical applications, and this is an urgent problem that needs to be resolved for base editing systems. The recently discovered phage-derived proteins, anti-CRISPRs, which can suppress the natural CRISPR nuclease activity, may be able to ameliorate the off-target effects of base editing systems. Here, we confirm for the first time that AcrIIA2, AcrIIA4, and AcrIIA5 efficiently inhibit base editing systems in human cells. In particular, AcrIIA5 has a significant inhibitory effect on all base editing variant systems tested in our study. We further show that the off-target effects of BE3 and ABE7.10 were significantly reduced in AcrIIA5 treated cells. This study suggests that AcrIIA5 should be widely used for the precise control of base editing and to thoroughly “shut off” nuclease activity of both CBE and ABE systems.

## 1. Introduction

The discovery of the CRISPR system, a widespread system used by bacteria for protection against potentially dangerous foreign DNA molecules [1,2,3], has led to a major shift in the fields of biotechnology and molecular biology, including genome editing and gene expression regulation in living cells and organisms [4,5]. While these technological innovations bring potential benefits to biology, medicine, and agriculture, concomitant problems arise over the largely irreversible outcomes generated by genome editing, including widely unexpected mutations and p53-dependent cell death [5,6,7,8].

In contrast to Cas9 nucleases, base editors do not create double-strand breaks (DSBs) and enable the precise installation of targeted point mutations in genomic DNA [9,10,11,12]. Cytosine base editors (CBEs) could be used for converting the C•G base pair to a T•A base pair, and adenine base editors (ABEs) convert the A•T base pair to a G•C base pair within a window of several nucleotides at a target site [9,10]. However, undesirable byproducts such as small insertions or deletions (indels) in cells [13,14], animals [15], plants [16], and other organisms [17,18,19] have been verified in recent studies. Moreover, unexpected off-target effects have been widely reported in genome-editing cells using both CBE and ABE systems [20,21,22]. To improve the efficiency and specificity, various base editor variants have been developed [23]. To a large extent, the efficiency and specificity of these base editors has been improved; however, the off-target effects cannot be solved fundamentally. Thus, the unexpected off-target effects will restrict the applications of base editors to precision gene therapy in the future [24].

To date, several BE3 and ABE variants have been developed to reduce the off-target effects in recent studies [25,26]. Moreover, the phage-encoded anti-CRISPR proteins, a very elegant product of the newly discovered evolutionary arms race between bacteria and phages, enable the inactivation of CRISPR systems in nature [27,28]. These CRISPR antagonists are diverse and widespread among phages for the large diversity of CRISPR/Cas systems [29,30]. Currently, forty-four unique families of anti-CRISPR proteins have been reported in roles against both class 1 and class 2 CRISPR/Cas systems [31]. Remarkably, of these inhibitors, AcrIIA2, AcrIIA4, and AcrIIA5 could be used to inhibit type II-A *Streptococcus pyogenes* Cas9 (SpyCas9)-mediated gene editing [28,32,33]. Furthermore, numerous scientists have engineered anti-CRISPR proteins for the optogenetic control of CRISPR/Cas9 [34], for gene editing in eukaryotic cells [35] and as a capture ligand for CRISPR/Cas9 detection using the anti-CRISPR protein of AcrIIA4 [36].

The BE3 and ABE7.10 base editors are based on the CRISPR/Cas9 system-directed cytidine or adenine deaminase enzymatic activity, respectively. In addition, it has been demonstrated that AcrllA4 inhibits CRISPR/Cas9 on- and off-target genome editing in human cells [37]. Therefore, we attempted to investigate the inhibitory effect of anti-CRISPR proteins on base editor system (CBE and ABE) in this study. Our results show that AcrIIA5 suppresses base editors and reduces their off-target effects in mammalian cells efficiently. Taken together, our study provides new insight into the precise genome editing of base editors and off-target events for gene therapy in the future.

## 2. Materials and Methods

### 2.1. Plasmids and Oligonucleotides

All the gRNA sequences listed (Appendix A) were synthesized by GENEWIZ Biotechnology, Ltd. (Suzhou, China) and then inserted into the pBluescriptSKII+ U6-sgRNA(F+E) empty plasmid (Addgene #74707) through a *BbsI* site. pCMV-BE3, pCMV-ABE7.10, pCMV_AncBE4max and pCMV_ABEmax were obtained from Addgene (Plasmid #73021, #102919, #112094 and #112095). NG-AncBE4max, NG-ABEmax, xCas9-AncBE4max and xCas9-ABEmax were kept in this lab.

pJH372, pJH373, pJH375, and pJH376 (mammalian expression of AcrIIA1, AcrIIA2, AcrIIA3, and AcrIIA4) were obtained from Addgene (Plasmid #86839, #86840, #86841, and #86842). pCMV-T7-hAcrVA1-NLS (sv40) (BPK5050), pCMV-T7-hAcrVA2.1-NLS (sv40) (BPK5059), pCMV-T7-hAcrVA2-NLS (sv40) (AAS2283), pCMV-T7-hAcrVA3.1-NLS (sv40) (RTW2624), and pCMV-T7-hAcrVA3-NLS (sv40) (BPK5077) were obtained from Addgene (Plasmid #115136, #115137, #115138, #115139, and #115140). Human codon-optimized AcrIIA5, AcrIIC1, AcrIIC2, and AcrIIC3 [28] anti-CRISPR sequences (Appendix A) were synthesized by GenScriptBiotechnology, Ltd. and then inserted into pcDNA3.1(+) expression vectors through *BamHI/EcoRI* sites.

### 2.2. Cell Culture and Transfection

HEK293T cells were cultured in DMEM (Gibco) supplemented with 10% fetal bovine serum (Gibco), 1% penicillin and streptomycin (Gibco), 1% GlutaMAX (Gibco), and 1% MEM non-essential amino acids at 37 °C with 5% CO_2_. HEK293T cells were seeded into 6-well plates and transfected at 70–85% confluency with 0.5 μg of plasmid encoding sgRNA and 1.5 μg of plasmid expressing the base editor using Lipofectamine 3000 (Invitrogen) according to the manufacturer’s recommendations. Genomic DNA was isolated 48 h after the transfection using a TIANamp Genomic DNA Kit (TIANGEN) for on-target sites by PCR and Sanger sequencing.

### 2.3. Quantification of Genome Editing Frequency

All on-target sites were amplified using a Q5 High-Fidelity PCR Kit (NEB #E0555S) and then subjected to Sanger sequencing. All primers used for base-editing analysis are listed in Appendix A. We estimated the base editing efficiency by EditR (http://baseeditr.com/) [38].

### 2.4. Targeted Deep Sequencing

On-target and potential off-target sites were amplified with a Q5 High-Fidelity PCR Kit (NEB #E0555S) for deep sequencing library generation. The libraries were sequenced using Illumina MiniSeq with paired-end sequencing systems by Sangon Biotech (Shanghai, China). Base editing frequencies indicate the frequencies of modified target sites with at least one edit within the editing window (position 4–8). The source data of base editing frequencies are provided in Appendix A.

### 2.5. Statistical Analysis

All data were presented as the mean ± SD (at least three biologically independent experiments) and analyzed with unpaired two-tailed Student’s t-tests using GraphPadprism software 8.0.2. *p* < 0.05 was considered significantly different. * *p* < 0.05, ** *p* < 0.01, *** *p* < 0.001, **** *p* < 0.0001.

### 2.6. Data Availability

High-throughput sequencing reads were deposited in the NCBI Sequence Read Archive under PRJNA647636.

## 3. Results

### 3.1. AcrIIA2, AcrIIA4, and AcrIIA5 Can Inhibit DNA Base Editing in Mammalian Cells

To investigate the potential of Acrs as inhibitors for base editors, we designed multiple targeting sites on human endogenous genes to test the base editing efficiency of BE3 and ABE7.10 (Appendix A). We co-transfected HEK293T cells with base editor plasmids and guide RNA (gRNA) and isolated the genomic DNA after 48 h. Then, we quantified the base editing efficiency using PCR and EditR (http://baseeditr.com/). As expected, the results showed the 6.4–19.2% C•G-to-T•A conversion at seven tested sites (Figure 1a and Appendix A) using BE3 and the 12.1–14% A•T-to-G•C conversion at two tested sites using ABE7.10 (Figure 1b and Appendix A) in HEK293T cells.

To test the inhibitory effect of anti-CRISPR proteins on BE3 and ABE7.10 (Figure 1c), the base editor plasmid, gRNA plasmid and different inhibitors plasmids (AcrIIC1, AcrIIC2, AcrIIC3, AcrIIA1, AcrIIA2, AcrIIA3, AcrIIA4, AcrIIA5, AcrIIVA1, AcrIIVA2, AcrIIVA2.1, AcrIIVA3, and AcrIIVA3.1) were transfected together into HEK293T cells. As shown in Figure 1d,e and Appendix A, AcrIIA2, AcrIIA4 and AcrIIA5 significantly reduced the base editing efficiency of BE3 and ABE7.10 whereas the other Acrs had no inhibitory effect.

### 3.2. AcrIIA5 Suppress the Activity of Base Editing at Different Ratios

Anti-CRISPR phages had been shown to overcome CRISPR/Cas immunity by cooperation [39,40], and our initial experiments showed that AcrIIA5 could suppress base editors efficiently. However, it has not been elucidated whether AcrIIA5 inhibits BE3 or ABE7.10 by a dose-dependent effect or not. Thus, we next tested the AcrIIA5 gene for its ability to inhibit base editing in human cells at different concentrations. The HEK293T cells were transiently transfected with plasmids expressing BE3/ABE7.10 and a sgRNA targeting endogenous gene in the absence or presence of vectors expressing human codon-optimized AcrIIA5 gene at ratios to the BE3 or ABE7.10/AcrIIA5 plasmid of 1:1, 1:3, and 1:5. After 48 h, we isolated genomic DNA and quantified the base editing efficiency using PCR and EditR. The results show that AcrIIA5 could suppress the activity of base editing at different ratios, while no significant differences in dose-dependent effects (Figure 2a,b, and Appendix A). Thus, a 1:1 ratio of BE3 or ABE7.10/AcrIIA5 plasmids was used in further experiments.

### 3.3. AcrIIA5 Inhibits Base Editor Variants

With the increasing applications of base editors, numerous variants have been developed recently [13,41]. Recent studies have shown that AcrIIA5 potently inhibits all Cas9 homologs by preventing DNA binding and leading to sgRNA cleavage [42], suggesting that AcrIIA5 could efficiently inhibit different base editor variant systems. However, it has not been elucidated whether AcrIIA5 is a broad inhibitor of the base editor systems or not. To investigate the potential of AcrIIA5 for other base editor variants, AncBE4max (the most efficient C•G-to-T•A conversion system [13]), ABEmax (the most efficient A•T-to-G•C conversion system), NG-AncBE4amx, NG-ABEamx, xCas9-AncBE4max, and xCas9-ABEmax were used in this study. As expected, AcrIIA5 exerted an efficient inhibitory effect on these base editor variants (Figure 3 and Appendix A). In conclusion, we confirmed that AcrIIA5 could efficiently inhibit different base editor variant systems.

### 3.4. AcrIIA5 Can Diminish Off-Target Effects

It is worth noting that limiting the duration of nuclease activity can decrease off-target activity [24]. Therefore, to diminish the off-target effects of base editors, we tried to deliver AcrIIA5 at 3 or 6 h after base editing using BE3 or ABE7.10 (Figure 4a). We examined the on-target base editing efficiencies and off-target effects of the *EMX1*, *FANCF*, *TYRO3* and *HBG* sgRNA (Appendix A), and then we performed targeted deep-sequencing and measured BE3 and ABE7.10-induced substitution frequencies in HEK293T cells (Appendix A). Our results showed 5.14–5.43% off-target effects at the *EMX1* site and 1.17–1.33% off-target effects at the *FANCF* site using BE3, whereas we recorded 2.71–3.07% off-target effects at *TYRO3* site and 1.73–1.83% off-target effects at *HBG* site using ABE7.10 (Figure 4b,c and Appendix A).

To determine whether AcrIIA5 could reduce off-target effects or not, the cells were transfected using BE3 or ABE7.10 together with the guides RNA, treated with AcrIIA5-encoding plasmid after 3 or 6 h. Then, the base editing efficiency and off-target effects were measured by targeted deep-sequencing. The results show that the off-target effects of BE3 and ABE7.10 were significantly reduced in AcrIIA5 treated cells, which were then compared with the control cells (Figure 4b,c). Thus, our results suggested that AcrIIA5 could significantly reduce the off-target events of the base editor (CBE and ABE).

## 4. Discussion

Since the majority of verified human genetic diseases are point mutations, base editing has the potential to correct these disease-related mutations [43,44]. However, recent studies reported that CBE could induce considerable off-target effects in both animals and plants [20,22]. Thus, the serious off-target effects greatly increase the safety risk of gene therapy in the future.

Numerous reports have described that AcrIIA2, AcrIIA4, and AcrIIA5 can inhibit SpyCas9 in prokaryotic and eukaryotic cells [32,33]. In this study, we illustrated, for the first time, that AcrIIA2, AcrIIA4, and AcrIIA5 could suppress base editors in HEK293T cells. In addition, two articles published in *Cells* concurrently reported that anti-CRISPR phages cooperate to overcome CRISPR/Cas immunity and that Acr-phage epidemiology depends on the initial phage density [39,40]. Thus, we studied the inhibitory effect at different concentrations, and we found that the inhibitory effect did not increase obviously with an increasing concentration because strong inhibition was achieved due to the initial concentration. Subsequently, we evaluated the inhibitory ability of AcrIIA5 on different base editors containing AncBE4max, ABEmax, NG-AncBE4amx, NG-ABEamx, xCas9-AncBE4max, and xCas9-ABEmax in human cells. Our results showed that AcrIIA5 could efficiently suppress multiple variants of base editor, suggesting the AcrIIA5 is a broad inhibitor of the base editor systems.

One of the important factors in determining the off-target effect is the intracellular concentration of the nuclease and the duration of nuclease activity [24]. In general, limiting the duration of nuclease activity can decrease off-target activity. Thus, we tried to add inhibitors at different time points after gene editing to reduce the off-target effects. Our results show significantly reduced off-target effects in the AcrIIA5-treated groups. Although these inhibitors partly sacrificed on-target editing (Appendix A), the inhibition effect on off-target editing is still worthy of recognition.

To summarize, we demonstrated that AcrIIA5 could efficiently inhibit various base editing systems and diminish the off-target effects of BE3 and ABE7.10 in human cells. Furthermore, the use of AcrIIA5 to control base editing events shows significant advantages to “shut down” nuclease activity, reducing the off-target effects of the base editors.

## Figures and Tables

**Figure 1 cells-09-01786-f001:**
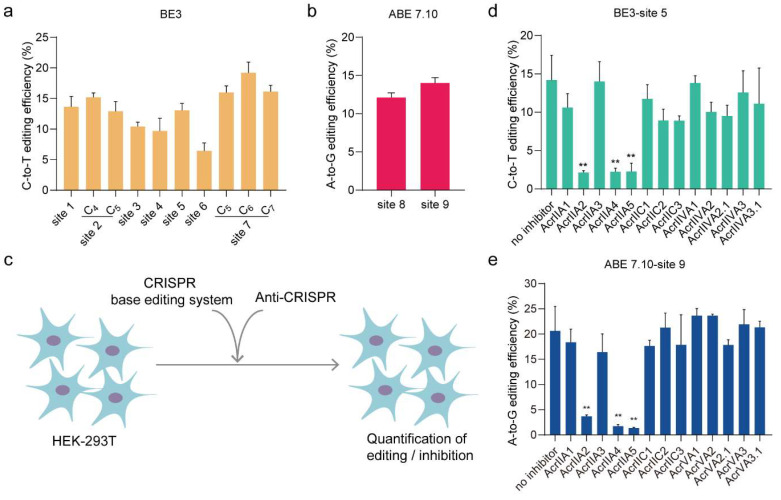
AcrIIA2, AcrIIA4, and AcrIIA5 inhibit BE3 and ABE7.10 in human cells. (**a**) C-to-T conversion efficiencies of BE3 at seven genomic sites. (**b**) A-to-G conversion efficiencies of ABE7.10 at two genomic sites. (**c**) Delivery of Anti-CRISPR plasmids inhibits BE3 or ABE7.10-mediated gene editing in human cells. HEK293T cells were transfected with BE3 or ABE7.10 plasmids and guide RNA (gRNA) and Anti-CRISPR plasmids. The base editing efficiency was estimated by EditR. (**d, e**) Anti-CRISPR genes were tested for their ability to inhibit BE3 (**d**) and ABE7.10 (**e**). All editing efficiencies shown represent the mean of three independent biological replicates. ** *p* < 0.01, by two-tailed Student’s *t*-test. Source data are provided in Appendix A.

**Figure 2 cells-09-01786-f002:**
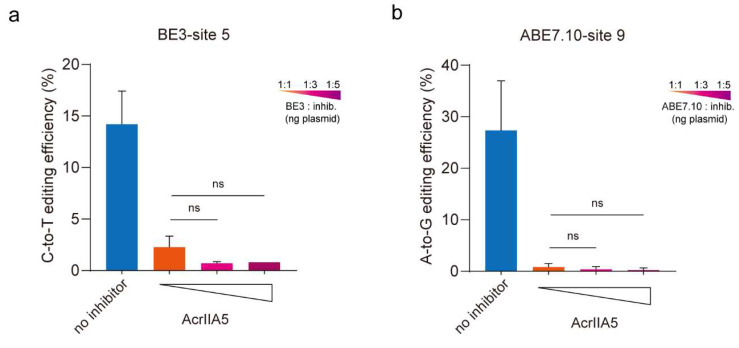
An increasing amount of AcrIIA5 plasmid (in ng) was tested for the ability to inhibit base editing. (**a**,**b**) AcrIIA5 plasmids were added at a ratio of 1:1 to 1:5 from left to right. All editing efficiencies shown represent the mean of three independent biological replicates. ns, no significance, by two-tailed Student’s *t*-test. Source data are provided in Appendix A.

**Figure 3 cells-09-01786-f003:**
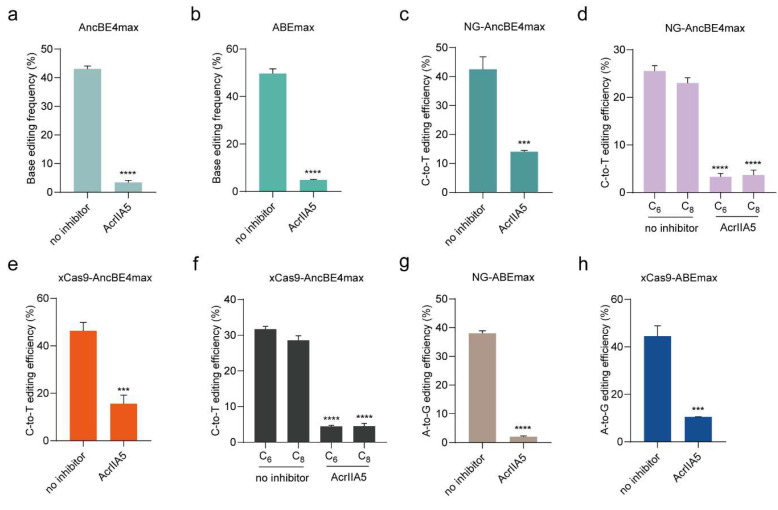
AcrIIA5 inhibits base editor variants in human cells. (**a**,**b**) AcrIIA5 was tested for its ability to inhibit AncBE4max at site 5 and ABEmax at site 9. (**c**–**f**) AcrIIA5 was tested for its ability to inhibit NG-AncBE4max (**c**,**d**) and xCas9-AncBE4max (**e**,**f**) at sites 10 and 11 containing an NGT protospacer adjacent motif (PAM). (**g**,**h**) AcrIIA5 was tested for its ability to inhibit NG-ABEmax and xCas9-ABEmax at site 9. All editing efficiencies shown represent the mean of three independent biological replicates. *** *p* < 0.001, **** *p* < 0.0001, by two-tailed Student’s *t*-test. Source data are provided in Appendix A.

**Figure 4 cells-09-01786-f004:**
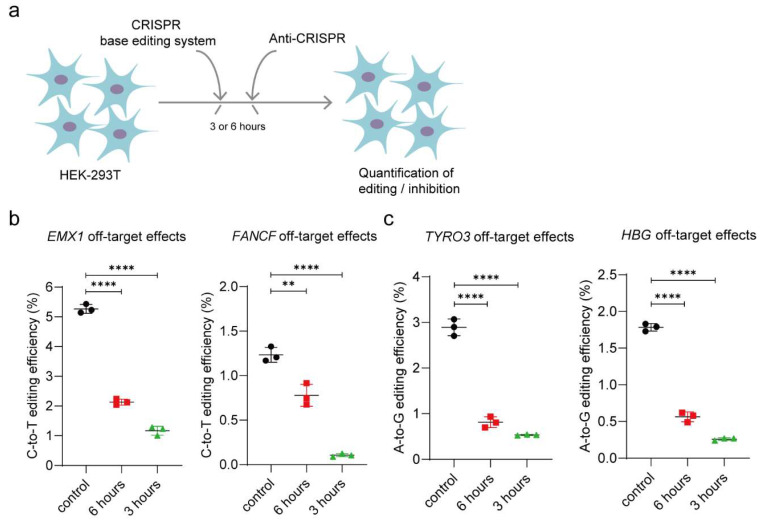
Decreased off-target effects of BE3 or ABE7.10 by AcrIIA5. (**a**) Delivery of AcrIIA5 after introduction of BE3 or ABE7.10 yields intermediate inhibition of base editor activity. HEK293T cells transfected with BE3 and ABE7.10 together with the guides RNA, and subsequently transfected again at 3 or 6 h with the plasmid encoding AcrIIA5. Base editing frequencies at off-target sites in HEK293T cells were measured by targeted deep sequencing. (**b**) AcrIIA5 can diminish off-target effects of BE3 system. (**c**) AcrIIA5 can diminish off-target effects of ABE7.10 system. ** *p* < 0.01, **** *p* < 0.0001, by two-tailed Student’s *t*-test. The source data of base editing frequencies are provided in Appendix A.

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
