# Peer review of "AcrIIA5 Suppresses Base Editors and Reduces Their Off-Target Effects"

_cells, 2020, doi:10.3390/cells9081786_

Round 1

Reviewer 1 Report

Liang et al. studied the effect of anti-CRISPR proteins on the on-target and off-target genome editing properties of BE3 and ABE7.10 base editors. They observed that the phage-derived AcrIIA2, AcrIIA4 and AcrIIA5 proteins could reduce the base editing efficiency of BE3 and ABE7.10 in HEK293T cells. In addition, AcrIIA5 inhibited base editing mediated by a set of base editor variant systems as well. This is an interesting manuscript; I suggest, however, a revision of the text to improve the wording of certain sentences before publication. 

In line 68 and 69, the meaning of the sentence starting with ...Therefore, we attempted... is not clear, the wording has to be amended.

In line 121, ...To test the inhibitory effect of BE3 and ABE... - change to:...To test the inhibitory effect of anti-CRISPR proteins on BE3 and ABE... 

In line 124 and 125 the sentence starting with ...As shown in... should be amended. I suggest: ...As shown in Fig. 1d-1e and Fig. S3-S6, AcrIIA2, AcrIIA4 and AcrIIA5 significantly reduced the base editing efficiency of BE3 and ABE7.10 whereas the other Acrs had no inhibitory effect.

In line 135 and 136, the sentence starting with ...Thus a 1:1... needs to be  improved. I suggest: ...Thus, a 1:1 ratio of BE3 or ABE7.10/AcrIIA5 plasmids was used in further experiments.

Line 163 to 166, the wording of the sentence starting with ...The results shown... should be changed because it is difficult to decipher the meaning of the current version.

Line 168 to 170, legend to Figure 4; I suggest to change the expressions "Timed delivery" and "Proper timing". I think that perhaps these axpressions do not convey exactly what the Authors ment. Similarly, in line 172, "proper time" should be changed, accordingly. 

In line 172, ...BE3-induced base editing...   - change to: ...BE3 and ABE7.10 induced base editing...

In line 172 to 174, the sentence starting with ...Notably... should be clearly formulated.

Line 198 and 199, the sentence starting with ...The quantitative...  should be amended.

One additional point regarding the Abstract: Line 27 and 28, ...In particular, AcrIIA5 has a greater inhibitory effect on all base editing variant systems. This statement in this form is not supported by the data described in Results, I think. I suggest: ...In particular, AcrIIA5 had a significant inhibitory effect on all base editing variant systems tested in our study.

I suggest the publication of the manuscripte after a major revision. 

Reviewer 2 Report

Dear editor

Thanks for the opportunity to review the present paper for cells

I am not an expert in the CRISPR/CRISPR modification field so I cannot give a full evaluation on the absolute novelty of authors findings.

Despite this, I believe the paper is overall technically and conceptually well done so I suggest publication upon an experimental revision

Heareafter are my comments.

In the present manuscript Liang et al. describe the effect of Anti-CRISPR proteins on the inhibition of CRISPR-based base editing in human cells. In particular they focused on AcrIIA5 which seems to be the most effective in inhibiting base editing and modulating off target effects. The work is overall well done but I suggest further experimental work to make it more clear and effective.

It is not clear to me how the experiment in figure 4 was performed. Were cells transfected with BE and ABE7.10 together with the guides RNA and subsequently transfected again at 3 and 6 hours with the plasmid encoding AcrIIA5 ? It should be clarified in figure legend and text. Also, authors should show the efficiency of base editing (how much it is affected) after transfection at 3 and 6 hours of AcrIIA5

Regarding again figure 4, could the authors show a lower time points (e.g. 1.5-2 hours) or a kinetics up to for example 12 hours (where I suppose they would find no effect ?) ?

Authors should use at least another additional cell line in their study (in some of the experiments)

Why do the authors focused only on AcrIIA5 ? it doesn’t look like AcrIIA2 and 4 are less effective. Do they have other data with them regarding off target limitation ?

All the best

Reviewer 3 Report

The authors briefly tested a combination of base editors and CRISPR inhibitors at a limited number of genomic loci to identify strategies to limit base editor off-target activity. The authors did identify 1-2 of such CRISPR inhibitors. The results are potentially interesting. However, the analysis is limited to 2 off-target sites and uses Sanger sequencing (a very low-resolution assay) to quantify off-targets. It is therefore not possible to generalize these findings, without expanding the analysis to several combination of target and off-target sites, and by employing for sensitive techniques such as targeted amplicon sequencing for the quantification of edited alleles. 

Round 2

Reviewer 1 Report

The Authors correctly addressed the points I raised. There are, however, some disturbing contradictions between the text of the manuscript and the Legends to the Supplementary Figures which should be double-checked, I think. In line 171 of the manuscript Figs. S27 (Sanger sequencing) and S30 (Base editing quantification data) are mentioned, correctly. Unfortunately, however, the Legend to supplementary figure S30 reads like this: "Figure S30. Base editing quantification data of Figure S28 Sanger sequencing."

Similarly, in line 172, S33 and the corresponding S36 are mentioned, correctly; unfortunately, however, the legend to S36 claims: "Base editing quantification data of Figure S34 Sanger sequencing" 

I think the legends to supplementary figures S30 and S36 should be corrected.

As a matter of fact, it seems to me that there is a recurrent error in the legends to the supplementary figiures S2, S4, S6, S8, S10, S12, S14, S16, S18, S20, S22, S24 and S26. E.g.: "Figure S2. Base editing quantification data of Figure S2 Sanger sequencing."   In this case a supplementary figure showing Sanger sequencing has the same number as the figure showing the base editing quantification data of the sequencing.  The correct legend would be: Figure S2. Base editing quantification data of Figure S1 Sanger sequencing.

The Legends to supplementary figures S31, S32, S37 and S38 should also be corrected in my opinion. E.g.: "Figure S38. Base editing quantification data of Figure S36 Sanger sequencing" - this is icertainly incorrect because S36 shows base editing quantification, not Sanger sequencing.

Line 171, ... using BE3. And...  - change to: ...using BE3, whereas we recorded...

I suggest a minor revision of the manuscript.

Reviewer 2 Report

I believe the manuscript is overall well done and I understand the points raised by the authors and the difficulties they are currenbtly facing due to shutting down of their laboratories for the spreading of the coronavirus infection in china

With this in mind i suggest acceptance of the present revised version

Author Response

Thank you very much for your helpful advices.

Reviewer 3 Report

Dear authors, 

I strongly believe that additional experiments are required to support the conclusions claimed in the manuscript, namely: testing targeting/off-targeting efficiencies at additional genomic loci; profiling of targeting/off-targeting using a more quantitative assay i.e. targeted sequencing (NGS).

I do understand that due to the current situation in China, it is not possible to perform additional experiments at this time. I do understand the sadness and frustration associated with the events and I empathize with the authors. However, I believe that experiments are necessary and I don't consider correct to recommend the editors to accept the paper at its current status. I will suggest the editor to provide an open deadline to allow resubmission of the manuscript without any stringent deadline.

Round 3

Reviewer 3 Report

Thanks for your effort on generating supporting data by amplicon sequencing. The method section is very poor with respect of library preparation and data processing. The authors should be transparent on data analysis and describe their pipeline in detail.

Other that that, I have no additional observations.
